# Transport Metabolons and Acid/Base Balance in Tumor Cells

**DOI:** 10.3390/cancers12040899

**Published:** 2020-04-07

**Authors:** Holger M. Becker, Joachim W. Deitmer

**Affiliations:** 1Institute of Physiological Chemistry, University of Veterinary Medicine Hannover, D-30559 Hannover, Germany; 2Department of Biology, University of Kaiserslautern, D-67653 Kaiserslautern, Germany; deitmer@biologie.uni-kl.de

**Keywords:** carbonic anhydrase, proton antenna, tumor metabolism, tumor cell migration

## Abstract

Solid tumors are metabolically highly active tissues, which produce large amounts of acid. The acid/base balance in tumor cells is regulated by the concerted interplay between a variety of membrane transporters and carbonic anhydrases (CAs), which cooperate to produce an alkaline intracellular, and an acidic extracellular, environment, in which cancer cells can outcompete their adjacent host cells. Many acid/base transporters form a structural and functional complex with CAs, coined “transport metabolon”. Transport metabolons with bicarbonate transporters require the binding of CA to the transporter and CA enzymatic activity. In cancer cells, these bicarbonate transport metabolons have been attributed a role in pH regulation and cell migration. Another type of transport metabolon is formed between CAs and monocarboxylate transporters, which mediate proton-coupled lactate transport across the cell membrane. In this complex, CAs function as “proton antenna” for the transporter, which mediate the rapid exchange of protons between the transporter and the surroundings. These transport metabolons do not require CA catalytic activity, and support the rapid efflux of lactate and protons from hypoxic cancer cells to allow sustained glycolytic activity and cell proliferation. Due to their prominent role in tumor acid/base regulation and metabolism, transport metabolons might be promising drug targets for new approaches in cancer therapy.

## 1. Introduction

The activity of many proteins is strongly influenced by their state of protonation. Therefore, intracellular and extracellular pH have to be tightly controlled to allow normal cell function. In healthy tissue, intracellular pH (pH_i_) is set to around neutral, while extracellular pH (pH_e_) is slightly alkaline [1,2]. In tumors, however, alterations in cell metabolism and the acid/base regulator machinery result in drastic changes in intracellular and extracellular pH, with a slightly alkaline pH_i_ and acidic pH_e_, which can drop down to values as low as 6.5 [3,4,5]. This extreme pH gradient seems to be a ubiquitous feature of malignant tumors, which appears already at an early stage of carcinogenesis, independent from the tumor’s origin and genetic background [6,7]. Alterations in intracellular and extracellular pH, in turn, trigger a variety of physiological responses, which drive tumor progression. The slight increase in pH_i_ supports cell proliferation and inhibits apoptosis [8,9,10,11]. Furthermore, a high pH_i_ activates cofilin, which supports cell migration by reorganization of the actin cytoskeleton [12,13]. Finally, the intracellular alkalinization enhances glycolysis, which results in an increased production of acid and exacerbates extracellular acidification. Extracellular acidification can kill adjacent host cells and suppresses an immune response by inhibiting T-cell activation and impaired chemotaxis [14,15]. Furthermore, the low pH_e_ supports degradation of the extracellular matrix (ECM) and thereby contributes to tumor cell migration and invasion [13,16].

pH regulation in tumor cells is governed by the concerted interplay between various acid/base transporters and carbonic anhydrases. The major pH regulator in cancer cells is the Na^+^/H^+^ exchanger 1 (NHE1). The expression of NHE1 is upregulated already in an early stage in tumorigenesis [6]. Oncogene-induced activation of NHE1 drives both intracellular alkalization and extracellular acidification, and was therefore suggested to play a key role in the malignant transformation of solid tumors [6,17]. Furthermore, NHE1, which accumulates in the leading edge of migrating cells, has been attributed a central role in cell migration and invasion [18,19,20,21]. However, other NHE isoforms, like NHE 6 and NHE9, have also been attributed important roles in tumor pH regulation, carcinogensis, and the development of chemoresistance [22,23,24]. NHE6 was shown to relocate from the endosomes to the plasma membrane, triggering endosome hyperacidification and chemoresistance in human MDA-MB-231 and HT-1080 cells [24].

Proton extrusion from cancer cells is further mediated by the monocarboxylate transporters MCT1 and MCT4, which are upregulated in a variety of tumor types [25,26,27,28]. MCTs transport lactate, but also other metabolites like pyruvate, ketone bodies, or branched-chain ketoacids, in cotransport with H^+^ across the plasma membrane [29,30,31,32]. Lactate-coupled proton extrusion from highly glycolytic cancer cells can exacerbate extracellular acidification and contribute to malignant transformation [33]. 

Besides the constant extrusion of protons, intracellular pH is regulated by the Na^+^/HCO_3_^−^ cotransporters NBCe1 and NBCn1. The electroneutral NBCn1 transports Na^+^ and HCO_3_^−^ in a 1:1 stoichiometry, while the electrogenic NBCe1 operates at a 1 Na^+^/2 HCO_3_^−^ or 1 Na^+^/3 HCO_3_^−^ stoichiometry, depending on the phosphorylation state of the transporter [34,35,36]. Indeed, NBCs have been suggested to be the major acid extruders (HCO_3_^−^ importers) in some breast cancer cells [37,38]. During cell migration, NBCs contribute to local pH changes by importing HCO_3_^−^ at the cell’s leading edge [13,39]. In this compartment, NBCs cooperate with the Cl^−^/HCO_3_^−^ exchanger AE2, which extrudes HCO_3_^−^ in exchange for osmotically active Cl^−^ to support local cell swelling [13,39]. The function of the various bicarbonate transporters in tumor acid/base regulation has been extensively discussed in several review articles [38,40,41].

Tumor pH regulation is further supported by carbonic anhydrases (CAs), which catalyze the reversible hydration of CO_2_ to HCO_3_^−^ and H^+^. Out of the 12 catalytically active CA isoforms expressed in humans, CAIX and CAXII have been attributed a distinct role in tumor pH regulation. In healthy tissue, the expression of CAIX is mainly restricted to the stomach and intestine. In cancer cells, however, the expression of CAIX is often upregulated and correlates with chemoresistance and poor clinical outcome [42,43,44]. The expression of CAIX is controlled by the hypoxia-inducible factor (HIF-1α), and therefore CAIX is primarily (but not exclusively) found in hypoxic tumor regions. CAIX comprises an exofacial catalytic domain, which is tethered to the plasma membrane with a single transmembrane domain, and an N-terminal proteoglycane-like (PG) domain, which is unique to CAIX amongst the carbonic anhydrases. The PG domain, which is rich in glutamate and aspartate residues, was suggested to function as a proton buffer for the catalytic domain and contributes to the formation of focal adhesion contacts [45,46]. In tumor tissue, CAIX functions as a “pH-stat”, which stabilizes the extracellular pH to 6.8 [47]. Like CAIX, CAXII is often upregulated in tumor cells; however, the isoform is, as compared to CAIX, more abundant in healthy cells, too. In contrast to CAIX, expression of CAXII was correlated to both good and bad prognosis, depending on the tumor type [48,49]. However, CAXII seems to play a role in the development of multidrug resistance and was therefore suggested as a potential drug target to overcome chemoresistance [50,51].

Besides the extracellular CA isoforms IX and XII, different cytosolic CAs, like CAI and CAII, have been attributed a role in various tumor types. However, the function of cytosolic CAs for tumor development and progression has not been studied as extensively as for their exofacial counterparts. For a comprehensive review about the role of the different CA isoforms in tumor cells, see [52].

## 2. Acid/Base Transport Metabolons

Intracellular and extracellular carbonic anhydrases interact with acid/base transporters, including NHEs, NBCs, AEs, and MCTs, to form a protein complex coined “transport metabolon”. A transport metabolon was defined as a “temporary, structural-functional, supramolecular complex of sequential metabolic enzymes and cellular structural elements, in which metabolites are passed from one active site to another, without complete equilibration with the bulk cellular fluids (channelling)” [53,54,55].

The best studied CA isoform that interacts with acid/base transporters is the intracellular CAII. CAII was shown to bind to an acidic motive (L^886^DADD) in the C-terminal tail of the Cl^−^/HCO_3_^−^ exchanger AE1 (band3) [56,57,58] (which is also expressed in different types of cancer, including gastric, colonic, and esophageal cancer [59,60]). Inhibition of CAII catalytic activity or overexpression of the catalytically inactive mutant CAII-V143Y reduced the transport activity of AE1, expressed in human embryonic kidney 293 (HEK-293) cells [61]. Based on these results, it was suggested that CAII, which is directly bound to the transporter, could locally provide HCO_3_^−^ to the exchanger. Thereby, CAII counteracts local depletion of HCO_3_^−^ and stabilizes the substrate pool for the transporter (Figure 1A).

CAII was also found to bind to, and facilitate the transport activity of, the Na^+^/HCO_3_^−^ cotransporters NBCe1 and NBCn1 [62,63,64,65,66], as well as the Na^+^/H^+^ exchangers NHE1 and NHE3 [67,68,69]. CAII-mediated facilitation of NBC and NHE transport activity requires direct binding of the enzyme to an acidic cluster in the transporters’ C-terminal tail, as well as CAII catalytic activity. In the case of NBC, CAII would either provide or remove HCO_3_^−^ to/from the transporter, while in the case NHE, CAII would provide H^+^ (Figure 1B,C).

The activity of HCO_3_^−^ transporters is also facilitated by direct and functional interaction with extracellular CA isoforms. CAIV was shown to bind to the fourth extracellular loop of both AE1 and NBCe1 [70,71]. The binding of CAIV as well as CAIV enzymatic activity is mandatory for the CAIV-mediated increase in HCO_3_^−^ transport. Thereby, CAIV would form the extracellular part of the transport metabolon, which provides or removes HCO_3_^−^ to/from the transporter to stabilize the HCO_3_^−^ gradient for maximum transport function (Figure 1A,B). In the case of NHE, CAIV would remove protons from the transporter to stabilize the H^+^ gradient (Figure 1C). The transport activity of Cl^−^/HCO_3_^−^ exchangers is not only facilitated by CAIV but also by CAIX, the catalytic domain of which binds to AE1, AE2, and AE3 to support Cl^−^/HCO_3_^−^ transport activity [72,73]. An overview of the various types of acid/base transport metabolons described so far is given in Table 1.

Even though there is accumulating evidence for the physical and functional interaction between acid/base transporters and CAs, both in vitro and in intact cells, several studies have questioned the existence of bicarbonate transport metabolons, both in respect to the physical as well as functional interaction between CAs and transport proteins [102,103,104,105]. For a comprehensive review on bicarbonate transport metabolons, including the criticism on this concept, we recommend the reviews [55,106,107,108].

A different type of transport metabolon is formed between intracellular and extracellular CAs and MCTs. In marked contrast to the transport metabolons formed with HCO_3_^−^ transporters, the facilitation of MCT activity is independent of CA catalytic activity [86,87,88,99]. In fact, the CAs seem to function as a “proton antenna”, which moves H^+^ between the MCT transporter pore and surrounding protonatable residues [88]. Proton-collecting antennae have been described for different H^+^-transporters, including cytochrome c oxidase and bacteriorhodopsin [109,110]. A proton-collecting antenna is comprised of several acidic glutamate and aspartate residues, which function as “H^+^ collectors”, and histidine residues, which function as “H^+^ retainers”. The antenna accelerates the protonation rate of functional groups and creates a “proton reservoir” on the protein surface [111]. CAII facilitates the transport activity of MCT1 and MCT4, when heterologously expressed in *Xenopus* oocytes [86,99]. Pharmacological inhibition of CAII catalytic activity as well as co-expression of the catalytically inactive mutant CAII-V143Y did not suppress CAII-mediated augmentation of MCT transport activity, indicating that the CAII-MCT1/4 metabolon functions independently of the CAII catalytic activity [86,87,88,99]. However, CAII-mediated facilitation of MCT1/4 activity requires direct binding of the enzyme to a cluster of three glutamic acid residues in the transporters’ C-terminal tail [89,98]. Interestingly, the transport activity of MCT2, which does not feature a CAII binding site, is not facilitated by co-expression with CAII [97]. However, integration of a CAII binding site into the MCT2 C-terminal tail allows CAII to facilitate MCT2 transport activity when heterologously expressed in *Xenopus* oocytes [98]. The binding of CAII to the MCT C-terminal tail is mediated by CAII-His64 [92]. Interestingly, His64 forms the central residue of the CAII intramolecular H^+^ shuttle, which is crucial for CAII to achieve high catalytic rates [112]. However, CAII-His64 does not participate in H^+^ shuttling between the transporter and enzyme [92]. This H^+^ shuttle seems to be mediated by the amino acids Glu69 and Asp72, which have been suggested to form a part of the CAII proton-collecting antenna [92,113]. Based on these results, it was suggested that CAII, which is directly bound to MCT1/4, facilitates parts of its proton-collecting apparatus to quickly move H^+^ between the transporter and the surrounding protonatable residues near the inner face of the cell membrane. Thereby, CAII functions as a proton antenna for the MCT, and counteracts the formation of H^+^ nanodomains (local accumulation or depletion of H^+^) around the transporter pore, which would impair proton-coupled lactate flux (Figure 2) [88,92,93,100].

MCT transport activity is not only facilitated by intracellular CAII but also by the extracellular isoforms CAIV and CAIX [90,91,94,95,96,97]. Up to now, the MCT-CAIV transport metabolon has only been investigated in *Xenopus* oocytes [90,94,97], while CAIX was shown to interact with MCTs both in the *Xenopus* oocyte expression system and in human breast cancer cell lines [91,95,96]. As already observed for CAII, the facilitation of MCT transport activity by extracellular CAs is independent of the enzymes’ catalytic activity [91,97] but requires close association of the CA to the transporter [94,96]. Interestingly, CAIV and CAIX do not directly bind to the transporter but to the first globular (Ig1) domain of the MCT chaperons CD147 (MCT1 and MCT4) and GP70 (MCT2) [94,97]. In both carbonic anhydrases, this binding is mediated by a histidine residue that is analogue to His64 in CAII (CAIV-His88 and CAIX-His200) [91,94,96]. Proton shuttling between MCTs and CAIX seems to be mediated by the CAIX PG domain, which features 18 glutamate and 8 aspartate residues. These acidic residues could function as proton antenna for the protein complex [96,114]. The proton antenna in CAIV is yet unidentified. Both CAIV and CAIX have been suggested to function as an extracellular H^+^ antenna for MCTs, which shuttles H^+^ between the transporter and surrounding protonatable residues, thereby counteracting the formation of extracellular H^+^ nanodomains (Figure 2) [91,94,96]. Since dissipation of the proton nanodomain on only one side of the membrane could exacerbate the accumulation or depletion of protons on the other side, due to increased MCT transport activity, and intracellular and extracellular CAs have to work in concert for efficient proton handling on both sides of the membrane to allow maximum MCT transport activity (Figure 2) [90,93].

Besides cancer cells, transport metabolons have also been suggested to operate in various cells and tissues, including erythrocytes (AE1-CAII) [56], kidney (NBCe1-CAII; NHE3-CAII) [63,68], gastric mucosal epithelium (AE2-CAIX) [72], heart muscle (NHE1-CAII; AE3-CAII) [69,115], and brain (MCT1-CAII; MCT-CAIV/CAXIV; AE3-CAIV/CAXIV) [89,116,117].

## 3. The Role of Transport Metabolons in Tumor Metabolism

Solid tumors, which often have to cope with acute local hypoxia, are highly glycolytic tissues, which produce large amounts of lactate and protons [118,119]. Proton-coupled lactate extrusion from cancer cells is primarily mediated by the monocarboxylate transporters MCT1 and MCT4 [25,26,27,28]. The constant efflux of acid into the pericellular space supports the formation of a hostile tumor microenvironment, in which tumor cells can outcompete their normal host cells [94]. Lactate production is increased under hypoxia, resulting in the need for enhanced lactate export capacity in cancer cells [91,120,121]. This increase in lactate transport could either be mediated by increased MCT expression levels or by the facilitation of MCT transport function through the formation of a transport metabolon [91,122,123]. Transport metabolons, formed between MCT1/MCT4 and CAIX, were found in tissue samples of human breast cancer patients but are absent in healthy breast tissue [96]. Interestingly, the number of MCT1/4–CAIX interactions, as shown by an in situ proximity ligation assay (PLA), increased with increasing tumor grade [96]. The increase in glycolysis, as found in higher grade tumors [124], appears to require a higher lactate efflux capacity in cells, which is achieved by an increasing number of transport metabolons [96]. Thereby, CAIX-facilitated H^+^/lactate efflux would enable sustained energy production in glycolytic cancer cells to allow continued cell proliferation and tumor progression. Increased glycolysis is often initiated by hypoxia [119,125]. In line with this, MCT1/4-CAIX transport metabolons were found in hypoxic MCF-7 and MDA-MB-231 breast cancer cells but not in normoxic cells [96]. Both breast cancer cell lines show an increase in glycolysis under hypoxia, which is accompanied by an increase in lactate transport capacity [91,96]. This hypoxia-induced increase in lactate transport is not mediated by increased expression of monocarboxylate transporters but by the non-catalytic function of CAIX, the expression of which is highly increased in MCF-7 and MDA-MB-231 cells under hypoxia [91]. Indeed, both knockdown of CAIX with siRNA, as well as application of an antibody against the CAIX PG domain blocked the hypoxia-induced increase in lactate flux, while inhibition of CA enzymatic activity with 6-ethoxy-2-benzothiazolesulfonamide (EZA) or an antibody directed against the CAIX catalytic domain did not affect lactate transport [91,95]. These data let to the conclusion that CAIX, the expression of which is increased under hypoxia, forms a non-catalytic transport metabolon with MCT1 and MCT4 to facilitate H^+^-coupled lactate efflux from glycolytic breast cancer cells. The rapid extrusion of lactate and H^+^ allows sustained glycolytic activity and cell proliferation (Figure 3). Indeed, knockout of CAIX as well as the application of an antibody against the CAIX PG domain, but not inhibition of CA enzymatic activity, resulted in a significant reduction in the proliferation of hypoxic MCF-7 and MDA-MB-231 cells [91,95]. These data were confirmed in the triple-negative breast cancer cell line UFH-001 [101]. Knockdown of CAIX with a CRISPR/Cas9 approach decreased the glycolytic proton efflux rate (GlycoPER), as measured by Seahorse analysis, in pseudo-hypoxic UFH-001 cells treated with the HIF1α-stabilizing agent desferrioxamine (DFO). Yet, isoform-specific inhibition of CAIX catalytic activity with three ureido-substituted benzene sulfonamides (USBs) did not affect proton efflux, indicating that CAIX catalytic activity is not required to facilitate H^+^-coupled lactate flux in cancer cells.

Experiments on *Xenopus* oocytes and mathematical modeling have shown that efficient lactate shuttling via MCTs requires a carbonic anhydrase on both sides of the plasma membrane [90,93]. Even though the intracellular isoform CAII is not cancer specific, it is upregulated in different tumor cells, including breast, lung, colorectal, gastrointestinal, and prostate cancer [126,127,128,129]. PLA assays revealed that CAII is closely colocalized with MCT1 in MCF-7 cells [92]. In these cells, knockdown of CAII resulted in a significant decrease in lactate transport capacity [92]. Even though the expression of CAII is not increased under hypoxia, CAII knockdown had a stronger effect on lactate transport in hypoxic than in normoxic cells. Indeed, the hypoxia-induced increase in lactate transport, which is mediated by CAIX [91], was completely abolished in the absence of CAII. These results indicate that extracellular CAIX requires an intracellular counterpart to facilitate lactate transport from cancer cells. This is in line with previous findings that enhanced proton shuttling by a carbonic anhydrase on only one side of the plasma membrane creates a proton nanodomain on the opposing side, which in turn decelerates proton-coupled lactate flux [90,93]. During the efflux of lactate, CAII would collect protons from the intracellular face of the plasma membrane and shuttle them to the transporter pore. On the exofacial side, CAIX would remove the protons from the transporter and shuttle them to surrounding protonatable sites at the membrane. By this “push-and-pull” principle, intracellular CAII and extracellular CAIX would cooperate to ensure efficient efflux of lactate and protons from hypoxic cells to allow sustained glycolytic energy production (Figure 3). The removal of CA on either side of the membrane would lead to the formation of a proton domain around the MCT on this side and would therefore decrease MCT transport activity, which would lead to cytosolic accumulation of lactate and protons and ultimately to decreased glycolysis and cell proliferation. Indeed, this interpretation is supported by the finding that knockdown of either CAII or CAIX decreased cell proliferation in hypoxic MCF-7 breast cancer cells. For a comprehensive review about MCT transport metabolons in cancer cells, see also [130].

## 4. The Role of Transport Metabolons in Tumor pH Regulation and Cell Migration

pH regulation in cancer cells is governed by the concerted interplay between various acid/base transporters, which mediate the ion-coupled transport of H^+^ or HCO_3_^−^ across the cell membrane. Besides regulating cellular pH, many of these transporters have been attributed a central function in the initiation and regulation of cell migration and invasion [13]. Tumor cell migration is a multistep process, which requires the degradation of the extracellular matrix (ECM) by extrusion of acid and proteolytic enzymes, reorganization of the actin cytoskeleton, and coordinated formation and release of focal adhesion contacts between the cell and ECM [13]. All these processes are pH dependent and are therefore controlled by the regulatory machinery of the cell’s acid/base-status. The major pH regulator in cell migration is the Na^+^/H^+^ exchanger NHE1, which is redirected to the cell’s leading edge during the onset of migration and accumulates in the lamellipodia of migrating cells [13,18,131]. NHE1-driven proton extrusion at the protruding front leads to the formation of a pH gradient along the cell, with a low pH_e_ and a high pH_i_. Extracellular acidification facilitates degradation of the ECM and supports the formation of focal adhesion contacts via the activation of integrins [132,133,134]. Intracellular alkalinization promotes the activity of cofilin [12], which produces free barbed ends in the actin cytoskeleton, thereby inducing actin branching and the growth of actin filaments into the protruding front, which pushes forward the leading edge. NHE1 also organizes the shape of the cytoskeleton during migration by directly acting as a plasma membrane anchor for actin filaments [135]. Furthermore, NHE1-mediated Na^+^ influx drives osmotic swelling by water influx via aquaporins (Figure 4) [13,136,137]. pH regulation in the lamellipodia is further supported by Na^+^/HCO_3_^−^ cotransporters (either NBCe1 or NBCn1), which are found in the leading edge of migrating cells, where the Na^+^-coupled import of HCO_3_^−^ via NBCs contributes to the formation of an acidic pH_e_ and alkaline pH_i_ [39,138]. Another acid/base transporter, which is found in the lamellipodia of migrating tumor cells, is the Cl^−^/HCO_3_^−^ exchanger AE2 [131]. Even though AE2-mediated transport of HCO_3_^−^ could contribute to intracellular pH regulation, the major role of AE2 in cell migration seems to be the facilitation of isosmotic cell swelling. Like NHE1 and NBC, AE2 accumulates at the leading edge of migrating cells [131], where it mediates the import of osmotically active Cl^−^ for the exchange of HCO_3_^−^. In this respect, AE2 could functionally cooperate with NBCs. A fraction of the HCO_3_^−^, which is imported by NBCs in cotransport with Na^+^, is again extruded by AE2 for the exchange of Cl^−^. This “HCO_3_^−^ short circuit” results in the net uptake of osmotically active Na^+^ and Cl^−^ and promotes water uptake via aquaporins and local swelling of the lamellipodia (Figure 4) [13,39,131]. In the thin lemellipodia, which do not contain mitochondria, energy is mainly produced by glycolysis, which generates H^+^ and lactate. Both lactate and H^+^ are extruded by MCT4, which has been shown to accumulate at the leading edge of migrating cells and therefore contributes to extracellular acidification [139]. For a detailed review on the role of pH in cell migration, see [7,13,140,141].

Interestingly, the transporters mentioned above have been found to interact with CAIX in cancer cells. CAIX itself, which is redistributed to the lamellipodia of migrating cells [39,46], plays a central role in tumor cell migration and invasion [80,142]. CAIX catalytic activity is required for the generation of acidic pH_e_ nanodomains, which are mandatory for proteolytic cleavage of the ECM [80]. Furthermore, CAIX directly interacts with different actin-regulating proteins in the cytosol [80] and promotes focal cell adhesion through its extracellular PG domain [46].

First evidence for the formation of transport metabolons in migrating tumor cells was presented in 2012 [39]. The authors showed by PLA that CAIX is closely colocalized with NBCe1 in the lamellipodia of hypoxic A549 adenocarcinomic human alveolar basal epithelial cells [39]. The same group also confirmed the colocalization of CAIX with NBCe1 in the invadopodia of hypoxic HT1080 fibrosarcoma cells [80]. Knockdown of the CAIX protein, as well as pharmacological inhibition of CAIX enzymatic activity or overexpression of a catalytically inactive CAIX mutant, resulted in a decrease in cell migration as well as reduced formation of invadopodia and proteolytic cleavage of the ECM. The authors further showed that application of monoclonal antibodies against the CAIX catalytic domain or the PG domain inhibits the metastatic properties of TE-1 human squamous cell carcinoma cells in a quail chorioallantoic membrane invasion assay and of HT-2018 cells in a mouse lung colonization assay [80]. Based on these results, the authors concluded that CAIX forms a transport metabolon with NBCe1 in the leading edge of migrating and invading tumor cells. This transport metabolon facilitates local bicarbonate uptake into the cells and thereby contributes to the formation of local pH_i_ and pH_e_ gradients, which drive remodeling of the actin cytoskeleton and proteolytic cleavage of the ECM [35,80] (Figure 4). CAIX not only colocalizes with NBCe1 but also with the Cl^−^/HCO_3_^−^ exchanger AE2 as shown by PLA in the lamellipodia of hypoxic SiHa cervical cancer cells [39]. The AE2-CAIX transport metabolon could either contribute to apparent pH_i_ buffering or support cell swelling by the import of osmotically active Cl^−^ (Figure 4). However, it should be noted that both studies did not directly measure the transport activity of NBCe1 and AE2 in the presence and absence of CAIX. Therefore, it could not be excluded that CAIX is not functionally linked to NBCe1 and AE2, even though it is in close proximity of the transporters. 

CAIX was not only found to interact with bicarbonate transporters but also with the Na^+^/H^+^ exchanger NHE1 [81]. Co-immunoprecipitation of NHE1 and CAIX from hypoxic SiHa cells revealed direct binding between the two proteins. Interestingly, NHE1 not only co-precipitated with CAIX but also with the Na^+^/Ca^2+^ exchanger NCX1, which is closely colocalized to CAIX and NHE1 in hypoxic SiHa cells and RCC4 renal cell carcinoma cells, respectively. These data indicate that the three proteins form a transport metabolon to efficiently extrude protons from the cell. Thereby, CAIX would support the removal of H^+^ from the extracellular face of the NHE1 (possibly by functioning as a proton antenna for the transporter), while NCX1 would remove cytosolic Na^+^ by the exchange of Na^+^ for Ca^2+^ [81]. The proton extrusion facilitates extracellular acidosis and thereby contributes to the formation of a hostile tumor microenvironment. Indeed, gene silencing of NCX1 and inhibition of NCX1 activity with KB-R7943 (which also resulted in a loss of NCX1–CAIX interaction) resulted in intracellular acidification in SiHa cells and increased the number of necrotic cells in SiHa spheroids. These results indicate that the NCX1-CAIX-NHE1 transport metabolon plays a crucial role in tumor pH regulation and cell survival. Furthermore, pharmacological inhibition of NCX1 transport activity with KB-R7943 reduced the migration of SiHa and RCC4 cells, which hints to an additional function of the protein complex in cell migration [81].

Acid extrusion from the highly glycolytic lamellipodia and invadopodia is also supported by MCTs, the activity of which is increased by the non-catalytic function of CAIX, as already described in the previous chapter (Figure 4). The application of an antibody against the CAIX PG domain, but not against the CAIX catalytic domain, inhibited MCT transport activity and cell migration in hypoxic MDA-MB-231 and MCF-7 breast cancer cells, indicating that the MCT-CAIX transport metabolon might also play a role in cell migration [95]. However, since the CAIX PG domain was also attributed a direct function in cell migration, these findings need further investigation.

Taken together, it can be concluded that the transport metabolons, formed between CAIX and the membrane transporters NHE1, NBCe1, AE2, MCT4, and NCX1, play a crucial role in shaping intracellular and extracellular pH gradients at the protruding front of migrating tumor cells, thereby promoting tumor cell migration, invasion, and ultimately metastasis. Therefore, these transport metabolons are promising targets for novel drug therapies against cancer. However, further studies have to be performed to clarify the role of acid/base transport metabolons in tumor progression.

## 5. Transport Metabolons as Drug Targets in Tumor Therapy

Carbonic anhydrase inhibitors are used as pharmaceuticals in the treatment of different diseases, including glaucoma [143,144], epilepsy [145,146], and high-altitude illness [147,148]. Furthermore, several CA inhibitors are currently in clinical trial for the treatment of various types of solid tumors [149]. Transport metabolons, however, have not been subjected to preclinical investigation as drug targets for cancer therapy. Even though bicarbonate transport metabolons, per se, have not been investigated as drug targets, their dependence on carbonic anhydrase catalytic activity renders them susceptible to conventional CA inhibitors. Therefore, it should be kept in mind that part of the effect of CA inhibitors on tumor progression might be attributed to the inhibition of bicarbonate transport metabolons in cancer cells. Targeting acid/base transporters via their interaction with CAs could even be advantageous to direct inhibition of the transporters, since these transporters are not unique to cancer cells and therefore direct targeting of the transporter could cause severe side effects [150,151,152]. Transport metabolons, especially those formed with the cancer-related CA isoforms CAIX and CAXII, seem to be mostly restricted to cancer cells and should therefore be a more specific target as the transporters themselves. However, further studies are required to confirm this possibility.

In contrast to bicarbonate transport metabolons, transport metabolons formed between MCTs and CAs are independent from the CA catalytic activity [86,87,88]. Therefore, these transport metabolons should be resistant to classical CA inhibitors. Indeed, inhibition of CA catalytic activity with EZA had no effect on the lactate transport capacity or proliferation of hypoxic MCF-7 breast cancer cells [91]. In line with this, selective inhibition of CAIX with three different ureido-substituted benzene sulfonamides (USBs) did not alter the glycolytic rate in DFO-treated UFH-001 breast cancer cells, while CRISPR/Cas9-induced deletion of CAIX resulted in a significant reduction of glycolysis in the same cell type [101,153]. Since the transport metabolons formed between MCTs and CAIX are insensitive to the inhibition of CAIX catalytic activity, they have to be targeted either via the physical interaction between CAIX and the MCT chaperon CD147 or via the CAIX antennal function. Indeed, the application of an antibody against the CAIX PG domain, which was suggested to function as a proton antenna for the enzyme, reduced the lactate transport capacity and proliferation of hypoxic breast cancer cells [95]. The application of a monoclonal antibody directed against an epitope near the CAIX binding site in the Ig1 domain of CD147 resulted in the displacement of CAIX from the MCT1/4-CD147 complex, as shown by PLA in hypoxic MCF-7 cells [96]. This “metabolon disruption” decreased the lactate transport capacity in cancer cells, which led to a decrease in glycolytic activity and cell proliferation [96]. These findings provide a proof of concept that the transport metabolon formed between MCT1/4, CD147, and CAIX can be targeted to interfere with cancer cell metabolism and inhibit tumor progression.

## 6. Conclusions

Alterations in tumor metabolism and the machinery regulating the acid/base-status in cells result in a reversal in the pH gradient in solid tumors, with a slightly alkaline pH_i_ and an acidic pH_e_. This reversal in the pH gradient, which is considered a ubiquitous feature of solid tumors, triggers a variety of physiological responses, including increased glycolytic activity, enhanced cell proliferation, inhibition of apoptosis, immune escape, and cell migration. pH regulation in tumor cells is governed by the concerted interplay between various acid/base transporters, many of which form a physical and functional complex with CAs, coined “transport metabolon”. Transport metabolons formed between the cancer-associated CAIX and Cl^−^/HCO_3_^−^ exchangers (AEs), Na^+^/HCO_3_^−^ cotransporters (NBCs), and Na^+^/H^+^ exchangers (NHEs) require CAIX catalytic activity. These protein complexes, which are often found in the leading edge of migrating cells, have been attributed a role in tumor pH regulation and cell migration.

Transport metabolons formed between monocarboxylate transporters and CAIX/CAII are independent of the CA catalytic activity. In fact, CAIX and CAII seem to function as “proton antennae” for the transporter, which mediate the rapid exchange of H^+^ between the transporter pore and surrounding protonatable residues. Since MCTs mediate the coupled efflux of protons and lactate from cancer cells, these transport metabolons have been suggested to play an important role in the energy metabolism of hypoxic tumor cells.

Due to their manifold functions in cancer cells, transport metabolons represent promising drug targets for tumor therapy. Since bicarbonate transport metabolons depend on CA catalytic activity, they could be targeted with classical CA inhibitors, some of which are already in clinical trials. For the MCT1/4-CAIX metabolon, which operates independently of CAIX catalytic activity, new therapeutic drugs have to be developed, which could function as “metabolon disruptors” that dislocate CAIX from the transporter–chaperon complex to inhibit proton-coupled lactate release from the cell.

## Figures and Tables

**Figure 1 cancers-12-00899-f001:**
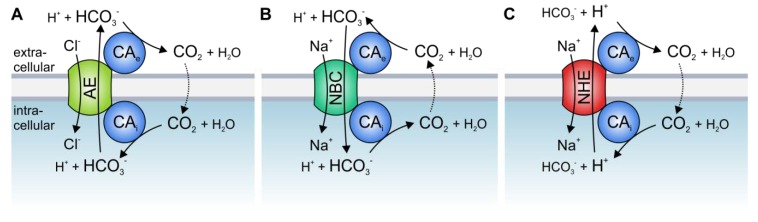
Hypothetical model of the interaction between acid/base transporters and carbonic anhydrases (CAs). Cytosolic CAs bind to the C-terminal tail of various isoforms of (**A**) Cl^−^/HCO_3_^−^ exchangers (AEs), (**B**) Na^+^/HCO_3_^−^ cotransporters (NBCs), and (**C**) Na^+^/H^+^ exchangers (NHEs), while exofacial CAs bind to an extracellular domain of the transporters. By catalyzing the reversible hydration of CO_2_ to HCO_3_^−^ and H^+^ in the direct vicinity of the transporter, CAs provide HCO_3_^−^ or H^+^ at the cis side and remove the HCO_3_^−^ or H^+^ on the trans side of the transporter.

**Figure 2 cancers-12-00899-f002:**
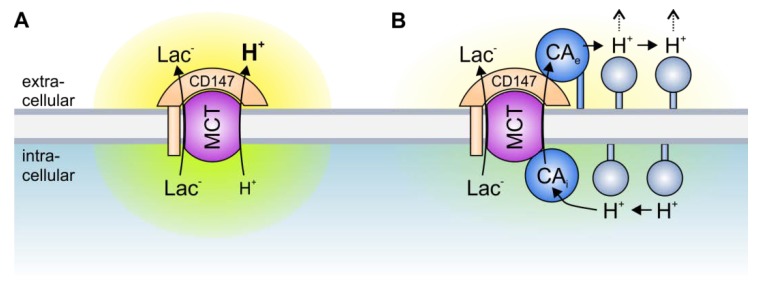
Transport metabolons with monocarboxylate transporters. (**A**) Due to the slow effective diffusion of protons (most protons are bound to bulky buffer molecules) MCT transport activity leads to the formation of proton nanodomains around the transporter pore. In the case of lactate and H^+^ efflux, H^+^ would locally deplete near the intracellular side of the membrane (green cloud) and locally accumulate near the extracellular side (yellow cloud). During lactate and H^+^ influx, H^+^ would deplete near the extracellular side and accumulate near the intracellular side of the membrane. These proton nanodomains would in turn decrease MCT transport activity. (**B**) During lactate and H^+^ efflux, intracellular CA (CA_i_), which is bound to the transporter’s C-terminal tail, provides H^+^ from surrounding protonatable residues to the transporter (thereby functioning as “proton-collecting antenna” for the transporter), while extracellular CA (CA_e_), which binds to the Ig1 domain of the MCT chaperon CD147, removes H^+^ from the transporter and shuttles them to surrounding protonatable residues (“proton-distributing antenna”). During lactate and H^+^ influx, CA_e_ would provide H^+^ to the transporter, while CA_i_ would remove the H^+^ from the transporter pore. By this “push and pull” mechanism, intracellular and extracellular CAs counteract the formation of proton nanodomains to allow rapid transport of lactate and H^+^ across the cell membrane.

**Figure 3 cancers-12-00899-f003:**
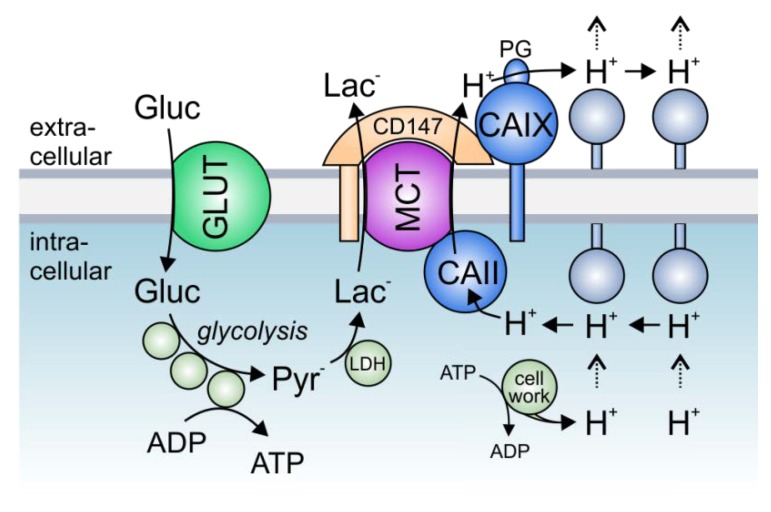
The role of lactate transport metabolons in cancer metabolism: Hypoxic tumors meet their energetic demand by glycolysis. Glucose, which is imported into the cell via glucose transporters (GLUTs), is metabolized to pyruvate, which in turn is converted to lactate by the enzyme lactate dehydrogenase (LDH). Lactate is exported in cotransport with protons (which are produced during metabolic activity) via monocarboxylate transporters (MCTs). In hypoxic cancer cells, MCT transport activity is facilitated by the interaction with intracellular CAII and extracellular CAIX, which directly bind to the transporters’ C-terminal tail and the MCT chaperon CD147, respectively. In this protein complex, CAII and CAIX function as a “proton antennae”, which facilitates the rapid exchange of H^+^ between the transporter pore and surrounding protonatable residues near the cell membrane. Thereby, CAII and CAIX drive the export of H^+^ and lactate from the cell to allow a high glycolytic rate and support cell proliferation.

**Figure 4 cancers-12-00899-f004:**
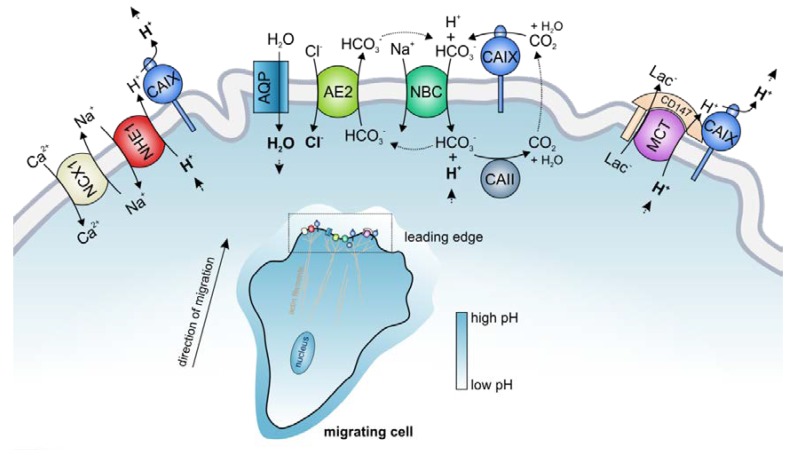
The function of transport metabolons in cancer cell migration. Tumor cell migration is a multistep process, which requires reversal of the pH gradient at the leading edge of the migrating cell. In most cancer cells, H^+^ extrusion is primarily mediated by the Na^+^/H^+^ exchanger NHE1 in exchange for Na^+^. NHE1 transport activity was suggested to be facilitated by CAIX, which removes H^+^ from the transporter pore. Protons are also removed by the monocarboxylate transporters MCT1 and MCT4 in cotransport with lactate. NHE1 transport activity could be further supported by the Na^+^/Ca^2+^ exchanger NCX1, which removes cytosolic Na^+^ by the exchange of Na^+^ for Ca^2+^. CAIX facilitates MCT1/4 transport activity by functioning as a proton antenna for the transporter, which removes H^+^ from the extracellular face of the transporter. The Na^+^/HCO_3_^−^ cotransporters NBCe1 and NBCn1 contribute to acid extrusion by Na^+^-coupled import of HCO_3_^−^. In the cytosol, HCO_3_^−^ binds H^+^, forming CO_2_, which can diffuse out of the cell. At the cell surface, CAIX hydrates CO_2_ again to HCO_3_^−^ and H^+^, and HCO_3_^−^ can be reimported into the cell. A fraction of the intracellular HCO_3_^−^ can again be exported by AE2 in exchange for Cl^−^. This “HCO_3_^−^ short circuit” results in cytosolic accumulation of osmotically active Na^+^ and Cl^−^, which facilitates water influx by aquaporins (AQPs) and supports local cell swelling. This complex interplay between acid/base transporters and carbonic anhydrases creates a pH gradient along the migrating cell with a low pH_e_ and a high pH_i_ at the protruding front, as shown in the inset picture.

**Table 1 cancers-12-00899-t001:** Overview of the known interactions between acid/base transporters and carbonic anhydrases. Combinations of CAs and transporters for which an interaction was demonstrated are indicated with ☑. CAs and transporters that have been shown not to interact are indicated with ⊠. Transport metabolons that have been shown in cancer cells are marked in red. For every CA, the left column indicates functional interaction. The right column indicates physical interaction. * CAIV and CAIX interact with MCT1, MCT2, and MCT4 via their chaperons CD (**cluster of difference**) 147 and GP70.

Transporter	Interacts with CA Isoform												Reference
	CA I		CA II		CA III		CA IV		CA IX		CA XIV		
AE1 (SLC4A1)		⊠	☑	☑			☑	☑	☑	☑			[39,56,57,58,61,70,72,74,75,76]
AE2 (SLC4A2)			☑	☑			☑		☑	☑			[39,57,61,70]
AE3 (SLC4A3)			☑				☑		☑	☑	☑	☑	[61,70,72,77]
DRA (SLC26A3)			☑	⊠									[70]
SLC26A6			☑	☑									[78]
SLC26A7			☑										[72]
NBCe1 (SLC4A4)	☑		☑	☑	☑		☑	☑	☑	☑			[39,62,63,64,65,66,71,73,79,80]
NBCn1 (SLC4A7)			☑	☑									[64]
NHE1 (SLC9A1)			☑	☑			☑		☑	☑			[81,67,79,82,83]
NHE3 (SLC9A3)			☑	☑									[68]
SNAT1 (SLC38A3)	☑		☑		☑		☑						[84,85]
MCT1 (SLC16A1)	⊠		☑	☑	⊠		☑	☑ *	☑	☑			[86,87,88,89,90,91,92,93,94,95,96]
MCT2 (SLC16A7)			⊠	⊠			☑	☑ *					[94,95,97,98]
MCT4 (SLC16A3)			☑	☑	⊠		☑	☑ *	☑	☑			[88,90,92,94,95,98,99,100,101]

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
