# Peer review of "Transport Metabolons and Acid/Base Balance in Tumor Cells"

_cancers, 2020, doi:10.3390/cancers12040899_

Round 1
Reviewer 1 Report
The article by Becker and Deitmer is a comprehensive review that covers the association between ion transporters and carbonic anhydrases. Tumor cells display a high capacity to extrude intracellular protons generated by their active metabolism, where molecular complexes formed by carbonic anhydrases and transporters are key players and might be potential new pharmacological targets for treating different types of cancer.
The article is well written and the figures are fine. However, I missed a table summarizing the interaction between transporters and carbonic anhydrases.
I strongly recommend the acceptance of this article after addressing the suggestion raised above.
Author Response
Thank you very much for the positive comments.
We added a table summarizing the various direct and functional interactions between acid/base transporters and carbonic anhydrases (Table 1).
Reviewer 2 Report
It is well designed to explain the molecular interaction for pH regulation. It is acceptable.
Author Response
Thanks a lot for this positive comment.
Reviewer 3 Report
In this review, Becker et al. have summarised evidence regarding the function of transport metabolons with acid/base transporters.
The review is largely well written and may be useful for some readers, although similar reviews have been published before and it contains information about research not carried out on tumour cells. In general, the authors should be more critical about some of the research they report, highlighting their limitations and the need for further studies.
More specific points include:
P1, line 35:
The authors should only cite original research, not reviews for statements like this. Ref 4 and 7 therefore are not appropriate.
P2, line 47:
“The major pH regulator in cancer cells is the Na+/H+ 47 exchanger NHE1”
Please include information on NHE6, which is also expressed in many tumours and has been shown to relocate from endosomes to plasma membrane under hypoxia recently.
P2, line 59:
More detailed information could be given on how NBcs achieve regulation of intracellular pH. In addition, the difference between NBCe1 and NBCn1 should be explained.
P2, line 93:
Is anything known about the role of AE1 in cancers? Or is AE1 only expressed in red blood cells and the kidney? Please include more information about the relevance of this metabolon in cancers.
In addition, CAII has been reported to mainly play a role in red blood cells. In general, it would be good to clarify in what tissue types this metabolons are found, and what cell lines the experiments were performed in.
P4 line 140:
‘Binding of CAII to the MCT C-terminal tail is mediated CAII-His64’ should it be ‘is mediated by CAII-His64’.
Figure 2:
the “push and pull” mechanism described by which intracellular and extracellular CAs counteract the formation of proton nanodomains to allow rapid transport of lactate and H+ across the cell membrane is not supported entirely by the literature. The authors explain earlier in the text that pharmacological inhibition of CAII catalytic activity as well as co-expression of the catalytically inactive mutant CAII-V143Y did not suppress CAII-mediated augmentation of MCT transport activity. Therefore the function of CAII binding to MCT cannot be to provide H+ directly to MCTs by its catalytic activity. If it is catalytically inactive, it will not provide protons. Please explain this paradox in more detail, otherwise the figure legend is misleading.
P5, line 180:
Many of the previously cited studies do not refer to cancer cells. Please make it more clear in the text on what tissues or cell lines these studies were carried out on.
P5, line 191:
‘Lactate production is increased under hypoxia, resulting in enhanced lactate export capacity in cancer cells.’ Please provide a reference for this statement.
P5, line 218:
‘tribble negative’ should be ‘triple negative’.
P7, line 275:
‘Intracellular alkalinisation promotes activity of cofilin’. Please add a reference here.
P9, line 339:
‘Based on these results the authors conclude that CAIX forms a transport metabolon with NBCe1’. Did this study also measure NBCe1 activity directly (for example via flux experiments)? Otherwise, CAIX and NBCe1 may be in close proximity, but CAIX may not functionally be linked to NBCe1. Were changes in pHi measured after CAIX knockdown? These could be due to other transporters present, for example NHE1. The authors should be more critical, highlight limitations of such studies and point to future research that should be done.
Author Response
Thank you very much for the constructive suggestions.
We considered all comments and made amendments to the manuscript as stated below:
P1, line 35:
The authors should only cite original research, not reviews for statements like this. Ref 4 and 7 therefore are not appropriate.
Ref 4 and 7 were removed from the text
P2, line 47:
“The major pH regulator in cancer cells is the Na+/H+ 47 exchanger NHE1”
Please include information on NHE6, which is also expressed in many tumours and has been shown to relocate from endosomes to plasma membrane under hypoxia recently.
We added some information about other NHEs in lines 52-56: “However, also other NHE isoforms, like NHE 6 and NHE9, have been attributed important roles in tumor pH regulation, carcinogensis and development of chemoresistance [22-24]. NHE6 was shown to relocate from the endosomes to the plasma membrane, triggering endosome hyperacidification and chemoresistance in human MDA-MB-231 and HT-1080 cells [24]. “
P2, line 59:
More detailed information could be given on how NBcs achieve regulation of intracellular pH. In addition, the difference between NBCe1 and NBCn1 should be explained.
We now explained the difference between NBCe1 and NBCn1 in lines 64-66: “The electroneutral NBCn1 transports Na+ and HCO3- in a 1:1 stoichiometry, while the electrogenic NBCe1 operates at a 1 Na+ / 2 HCO3- or 1 Na+ / 3 HCO3- stoichiometry, depending on the phosphorylation state of the transporter [34-36]. Indeed, NBCs have been suggested to be the major acid extruders (HCO3- importers) in some breast cancer cells [37, 38]. “
Since we feel that a detailed discussion of the role of NBCs in cancer pH regulation is bejond the scope of this review, we referred to other review articles on this topic in lines 71-72: “The function of the various bicarbonate transporters in tumor acid/base regulation has been extensively discussed in several review articles [38, 40, 41].”
P2, line 93:
Is anything known about the role of AE1 in cancers? Or is AE1 only expressed in red blood cells and the kidney? Please include more information about the relevance of this metabolon in cancers.
In addition, CAII has been reported to mainly play a role in red blood cells. In general, it would be good to clarify in what tissue types this metabolons are found, and what cell lines the experiments were performed in.
Expression of AE1 was found in gastric, colonic and esophageal cancer. We have now mentioned this in lines 103-104.
Transport metabolons, formed between bicarbonate transporters and CAII, have not yet been investigated in cancer cells until now. We have added a table to summarize the currently known transport metabolons and to point out which of these metabolons have already been identified in cancer cells.
The experiments on the AE1-CAII interaction were performed in HEK-293 cells (line 105)
We also added a short overview over which transport metabolons are found in which tissues (lines 209-212): “Besides cancer cells, transport metabolons have also been suggested to operate in various cells and tissues, including erythrocytes (AE1-CAII) [56], kidney (NBCe1-CAII; NHE3-CAII) [63, 68], gastric mucosal epithelium (AE2-CAIX) [72], heart muscle (NHE1-CAII; AE3-CAII) [69, 115] and brain (MCT1-CAII; MCT-CAIV/CAXIV; AE3-CAIV/CAXIV) [89, 116, 117]. “
P4 line 140:
‘Binding of CAII to the MCT C-terminal tail is mediated CAII-His64’ should it be ‘is mediated by CAII-His64’.
Error corrected
Figure 2:
the “push and pull” mechanism described by which intracellular and extracellular CAs counteract the formation of proton nanodomains to allow rapid transport of lactate and H+ across the cell membrane is not supported entirely by the literature. The authors explain earlier in the text that pharmacological inhibition of CAII catalytic activity as well as co-expression of the catalytically inactive mutant CAII-V143Y did not suppress CAII-mediated augmentation of MCT transport activity. Therefore the function of CAII binding to MCT cannot be to provide H+ directly to MCTs by its catalytic activity. If it is catalytically inactive, it will not provide protons. Please explain this paradox in more detail, otherwise the figure legend is misleading.
CAs can function as ‘proton antennae’ which mediate the rapid exchange of H+ between the MCT’s transporter pore and surrounding protonatable residues. This function is independent from the enzyme’s catalytic activity. We describe this mechanism now in more detail in lines 147-173: “In fact, the CAs seem to function as a ‘proton antenna’, which moves H+ between the MCT transporter pore and surrounding protonatable residues [88]. Proton-collecting antennae have been described for different H+-transporters, including cytochrome c oxidase and bacteriorhodopsin [109, 110]. A proton-collecting antenna is comprised of several acidic glutamate and aspartate residues, which function as ’H+ collectors’, and histidine residues, which function as ‘H+ retainers’. The antenna accelerates the protonation rate of functional groups and creates a ‘proton reservoir’ on the protein surface [111]. … “Based on these results it was suggested that CAII, which is directly bound to MCT1/4, facilitates parts of its proton-collecting apparatus to quickly move H+ between the transporter and the surrounding protonatable residues near the inner face of the cell membrane. Thereby, CAII functions as a proton antenna for the MCT, and counteracts the formation of H+ nanodomains (local accumulation or depletion of H+) around the transporter pore, which would impair proton-coupled lactate flux (Figure 2) [88, 92, 93, 100].“
Furthermore, we added the information that CAs function as proton antennae to the legend of Figure 2 (lines 183-186).
P5, line 180:
Many of the previously cited studies do not refer to cancer cells. Please make it more clear in the text on what tissues or cell lines these studies were carried out on.
We made this more clear now (lines 191-193): “Up to now, the MCT-CAIV transport metabolon has only been investigated in Xenopus oocytes [90, 94, 97], while CAIX was shown to interact with MCTs both in the Xenopus oocyte expression system and in human breast cancer cell lines [91, 95, 96].“
A detailed description of the experiments in cancer cells is then given in chapter 3.
P5, line 191:
‘Lactate production is increased under hypoxia, resulting in enhanced lactate export capacity in cancer cells.’ Please provide a reference for this statement.
References added
P5, line 218:
‘tribble negative’ should be ‘triple negative’.
Typo corrected
P7, line 275:
‘Intracellular alkalinisation promotes activity of cofilin’. Please add a reference here.
Reference added
P9, line 339:
‘Based on these results the authors conclude that CAIX forms a transport metabolon with NBCe1’. Did this study also measure NBCe1 activity directly (for example via flux experiments)? Otherwise, CAIX and NBCe1 may be in close proximity, but CAIX may not functionally be linked to NBCe1. Were changes in pHi measured after CAIX knockdown? These could be due to other transporters present, for example NHE1. The authors should be more critical, highlight limitations of such studies and point to future research that should be done
The reviewer has a point here. We now point out this limitation in lines 365-368: „ However, it should be noted that both studies did not directly measure transport activity of NBCe1 and AE2 in the presence and absence of CAIX. Therefore, it could not be excluded that CAIX is not functionally linked to NBCe1 and AE2, even though it is in close proximity to the transporters. “
Furthermore we added a sentence to the end of the chapter in lines 396-397: “However, further studies have to be performed to clarify the role of acid/base transport metabolons in tumor progression. “